# Diving into the Pleural Fluid: Liquid Biopsy for Metastatic Malignant Pleural Effusions

**DOI:** 10.3390/cancers13112798

**Published:** 2021-06-04

**Authors:** Maria Alba Sorolla, Anabel Sorolla, Eva Parisi, Antonieta Salud, José M. Porcel

**Affiliations:** 1Research Group of Cancer Biomarkers, Biomedical Research Institute of Lleida (IRBLleida), 25198 Lleida, Spain; msorolla@irblleida.cat (M.A.S.); asorolla@irblleida.cat (A.S.); eparisi@irblleida.cat (E.P.); masalud.lleida.ics@gencat.cat (A.S.); 2Department of Medical Oncology, Arnau de Vilanova University Hospital (HUAV), 25198 Lleida, Spain; 3Pleural Medicine Unit, Department of Internal Medicine, Arnau de Vilanova University Hospital (HUAV), 25198 Lleida, Spain

**Keywords:** liquid biopsy, pleural fluid, malignant pleural effusion, genomics, cytomics

## Abstract

**Simple Summary:**

Malignant pleural effusion is a common complication arising as the natural progression of many tumors, such as lung cancer. When this occurs, the common protocol consists of analyzing the pleural fluid for the presence of malignant cells. However, on many occasions no malignant cells are found despite a clear suspicion of cancer. Thus, the current diagnostic methodology is imperfect and more precise methods for the identification of malignancy are needed. Nonetheless, these methods are often invasive, which may be counterproductive, especially for patients with poor health condition. These concerns have made clinicians consider alternative non-invasive strategies to diagnose cancer using the generally abundant pleural fluid (e.g., liquid biopsy). Thus, a liquid sample can be analyzed for the presence of cancer footprints, such as circulating malignant cells and tumor nucleic acids. Herein, we review the literature for studies considering pleural fluid as a successful source of liquid biopsy.

**Abstract:**

Liquid biopsy is emerging as a promising non-invasive diagnostic tool for malignant pleural effusions (MPE) due to the low sensitivity of conventional pleural fluid (PF) cytological examination and the difficulty to obtain tissue biopsies, which are invasive and require procedural skills. Currently, liquid biopsy is increasingly being used for the detection of driver mutations in circulating tumor DNA (ctDNA) from plasma specimens to guide therapeutic interventions. Notably, malignant PF are richer than plasma in tumor-derived products with potential clinical usefulness, such as ctDNA, micro RNAs (miRNAs) and long non-coding RNAs (lncRNAs), and circulating tumor cells (CTC). Tumor-educated cell types, such as platelets and macrophages, have also been added to this diagnostic armamentarium. Herein, we will present an overview of the role of the preceding biomarkers, collectively known as liquid biopsy, in PF samples, as well as the main technical approaches used for their detection and quantitation, including a proper sample processing. Technical limitations of current platforms and future perspectives in the field will also be addressed. Using PF as liquid biopsy shows promise for use in current practice to facilitate the diagnosis and management of metastatic MPE.

## 1. Introduction

There are over 60 recognized causes of pleural effusions (PEs), among which cancer occupies a prominent place [1]. In fact, malignant pleural effusions (MPEs) are the leading cause of exudates in patients who undergo thoracentesis [2]. Cytological examination of the pleural fluid (PF), using stained smears and cell blocks preparation [3], is able to provide the diagnosis in around 55% of MPEs, thus leaving a significant proportion of undiagnosed cases [4]. Invasive procedures (e.g., pleuroscopic biopsy) may be necessary to uncover those MPEs with a false negative cytological result. In addition to the low sensitivity of PF cytology, the clinician is confronted with the need to characterize the molecular profile of the metastatic tumor to identify patients who can benefit from personalized therapies. Liquid biopsy applied to PFs, as a minimally invasive technology to capture fluid cancer biomarkers, may potentially overcome the limitations of conventional PF examination. It may allow not only to detect circulating tumor nucleic acids in cytologically-negative MPEs (an irrefutable proof of malignancy), but also to interrogate the tumor genome to detect driver and resistance molecular alterations with direct therapeutic implications. This narrative review addresses the current role of liquid biopsy approaches for MPE, with a particular focus on technological advances in the field.

## 2. Tumor-Derived Products in the Pleural Fluid

Tumors release a myriad of tumor-derived products into the PF as a result of their growth within the pleural cavity. These products are molecular indicators of tumor development and represent a source of detectable biomarkers, which are useful for diagnosis, prognosis, and therapeutical intervention. Essentially, they are nucleic acids such as ctDNA and non-coding RNA (ncRNA); cells entities that include CTCs and tumor-educated cells; and extracellular vesicles, also named as exosomes (Figure 1). The purpose of this article is to undergo an in-depth review about the reported presence of these specific molecules and cells on PF, as well as the identification of the major challenges and limitations for their detection and clinical implementation. We have excluded other molecules that are also present in PF such as proteins or metabolites, as they are not generally considered a part of liquid biopsy strategies.

### 2.1. Nucleic Acids

Essentially, liquid biopsy is based on the analysis of circulating nucleic acids. In 1953, Mandel and Metais reported for the first time the presence of cell free DNA (cfDNA) and RNA in human blood [5] and later, in 1977, individuals with cancer were found to exhibit high levels of cfDNA in their serum [6]. The increase in cfDNA in cancer patients is driven by ctDNA released from tumor cells. Detectable concentrations of ctDNA have been found in other biofluids, such as PF [7], urine [8], cerebrospinal fluid [9], and saliva [10]. Regarding circulating RNA, several species of RNA molecules, either free or packed in extracellular vesicles, have been reported to be present in plasma and serum, and used as liquid biopsy cancer biomarkers. These are mainly non-coding RNA (miRNA being the most known), lncRNAs, and transfer RNAs, among other species [11].

#### 2.1.1. Cell-Free DNA and Circulating Tumor DNA

CfDNA and ctDNA are passively released into the bloodstream by healthy and tumor cells, respectively, once cells die by necrosis or apoptosis. In addition, active release has also been hypothesized, particularly in tumor cells, as a strategy to promote malignancy on non-tumor cells [12]. Whilst typical concentrations of cfDNA in plasma of healthy individuals vary from 1 to 10 ng/mL, this concentration can substantially increase in certain conditions, such as acute cerebral infarction, exercise, transplantation, and infection [13]. Regarding PF, the quantification of cfDNA has demonstrated to be useful to differentiate exudative from transudative PEs [14]. A transudate is a plasma ultrafiltrate that usually results from a disequilibrium of the Starling forces (hydrostatic and oncotic pressures) in which a healthy pleura acts as a passive membrane [15]. Therefore, the small amounts of cfDNA detected in transudates derive from plasma rather than local production [14]. In contrast, an exudate implies inflammatory and cellular responses into the pleural cavity due to an increased microvessel permeability and/or lymphatic blockage [16]. The infiltration of leukocytes in infectious PE constitutes a source of cfDNA from dying cells [17], whereas the presence of a pleural tumor increases ctDNA concentrations. MPEs, which are typically exudative, showed a significant increase in cfDNA compared to transudative PEs. Considering only exudative etiologies, differences in cfDNA content between MPE and infectious PEs were also observed as significant, though less robust [14].

Besides quantitative differences on cfDNA content between PE etiologies, other authors highlight the measurement of parameters like the cfDNA fragmentation pattern, as it varies according to the cell death mechanism. Apoptotic cell death generates uniform small DNA fragmentation in plasma (~166 base pairs, bp), whereas necrotic cell death, occurring predominantly in tumor cells, generates a wide spectrum of longer fragments due to incomplete digestion of genomic DNA [18]. This phenomenon has been recently observed in PF, were cfDNA from lung associated-MPE contains much longer DNA fragments with sizes over 500 bp, compared to plasma [19]. Other authors emphasize the usefulness of the determination of DNA integrity index on PF, defined as the ratio between long and short cfDNA fragments, for diagnostic purposes. In one study, cytology and DNA integrity index taken together, offered 81 sensitivity and 87% specificity in distinguishing benign and malignant PE [20]. Interestingly, an elevated PF DNA integrity index had a positive predictive value of 81% in predicting cytology-negative MPE.

cfDNA content can be significantly increased in cancer patients, correlating with tumor volume size [21] and stage [22]. This is the reason why cfDNA, or more precisely ctDNA analysis, has become a promising tool for cancer monitoring: from the early detection of mutations on ctDNA to prognosis assessment, through the identification of minimal residual disease and prediction of disease recurrences [23]. First attempts for the identification of point mutations on ctDNA began in the early 1990s with the progress made by the Human Genome Project. In 1994, Vasioukhin et al. detected point mutations in *NRAS* in the leukemia patients’ blood [24]; and *KRAS* mutated sequences were identified in plasma and serum from patients with pancreatic adenocarcinoma [25]. Nevertheless, it was not until the past decade that researchers began to consider PF as a suitable source of detectable mutations on ctDNA. In this regard, the presence of a metastatic mass growing over the pleural membranes is undoubtedly an abundant source of ctDNA in PF. A study conducted by Porcel et al. determined that lung (37%) and breast cancer (16%) were the most common primary tumors causative of MPE, followed by hematological (10%) and gastrointestinal tumors (10%) [2]. Thus, it is not surprising that most studies on PF genomics are focused on lung cancer and, more specifically, non-small cell lung cancer (NSCLC).

One of the most studied genes in PF samples is Epidermal Growth Factor Receptor (*EGFR*), as it is the most common actionable target in NSCLC. Approximately, 50% of NSCLC cases in Asia [26] and around 10% in Caucasian European population [27] carry activating mutations on the tyrosine kinase domain of *EGFR*. Tyrosine kinase inhibitors (TKI) are the cornerstone of treatment for NSCLC, but resistances frequently occur, which have been attributed to secondary mutations in *EGFR*, being the T790M base substitution the most widely reported. Other relevant mutations in NSCLC involve *KRAS*, *MET*, and *PIK3CA*, with an overall prevalence of 23.0, 6.8, and 4.9%, respectively [28].

##### Genomic Strategies for Mutation Profiling in Pleural Fluid

In 2005, Huang et al. reported for the first time the detection of a specific *EGFR* mutation (exon 19: delE746—A750), using polymerase chain reaction (PCR) followed by direct DNA sequencing of the pleural cell pellet, in one NSCLC patient who responded to gefitinib treatment [29]. Another research showed that PF supernatants were comparable to tissue biopsy and superior to PF pellets (i.e., cell blocks) in successfully identifying *EGFR* point mutations on exons 19–21 by high-resolution melting and direct DNA sequencing in NSCLC patients samples [30]. Similarly, *EGFR* and *KRAS* mutations and *RET* and *ALK* rearrangements were identified by direct DNA sequencing in pre-amplified products from PF supernatants and pellets of 722 patients with metastatic lung adenocarcinoma with cytology-proven MPE [31]. Moreover, direct PCR sequencing was useful in identifying *EGFR* and *KRAS* mutations in formalin-fixed and paraffin-embedded (FFPE) samples from primary tumors and pleural metastasis, this latter including PF cell blocks and pleural biopsies of 37 lung adenocarcinoma patients [32]. Nevertheless, the authors claimed that there was a substantial discordance (16.2%) on *EGFR* status between the primary tumor and their corresponding metastasis. In these discordant cases, the responsiveness to *EGFR* TK (gefitinib or erlotinib) inhibitors was more likely to be correlated with *EGFR* mutations in metastatic lesions than in primary tumors. In another study, mutant-enrichment PCR-based on PCR-restriction fragment length polymorphism showed successful detection of *EGFR* mutations in both PF supernatants and pellets from 26 patients who were pathologically diagnosed with advanced NSCLC [33]. PCR and direct sequencing were applied to PF supernatants for the detection of mutations on *EGFR* (exons 18–21) in 11 out of 43 NSCLC patients. Interestingly, patients showing partial response or stable disease after treatment with gefitinib harbored *EGFR* mutations [34].

Another study successfully identified copy number variations (CNV) on *EGFR*, *NOTCH1*, *NF1*, *CCDN1*, and *MEN1*, among others; as well as point mutations (*PIK3CA* E542K) on ctDNA from ascites and PF supernatants of cancer patients by using genomic microarray hybridization [35]. Interestingly, ctDNA isolated from PF supernatants and extracellular vesicles (EV) presented 88 and 91% of concordance with tissue, respectively, regarding *EGFR* mutation detection by the peptide nucleic acid (PNA)-mediated PCR clamping method [36]. Moreover, ctDNA from EV seemed to be the best sample type for the detection of *EGFR* T790M mutations. These kind of mutations, which confer resistance to TKI, were detected in 72% of EV, whereas the detection rate in ctDNA from PF supernatants and cell blocks were 61 and 17%, respectively. By using the same methodology, Yang et al. detected 47 different mutations in the *EGFR* exons 18–24 in 72.5% of the 40 cytopathology-positive PF cell blocks examined. When looking at the clinical data, the study revealed that patients treated with TKI whose PF pellets carried *EGFR* mutations had better progression-free survival compared to those that were wild-type (7.33 versus 2.07 months, *p* = 0.032). In addition, the objective response rate was better in the EGFR-mutated patients compared to the non-mutated individuals (80.8% versus 10%, *p* < 0.001) [37]. Similarly, one additional investigation assessed *EGFR* status in 37 histologically diagnosed NSCLC patients with available tumor tissue, PF (supernatants and cell blocks), and serum [38]. It was found that there was a high concordance between PNA clamping methodology and direct PCR sequencing across all sample types. Nevertheless, the diagnostic performance of PE was higher using the PNA clamping method than direct PCR sequencing (89% sensitivity versus 100% specificity). Another study further supported the superiority of PNA clamping for the detection of *EGFR* mutation in PF cell blocks from lung adenocarcinoma patients with a low proportion of tumor cells [39].

A different methodology for the identification of mutations in the *EGFR* gene is the real-time quantitative PCR (RTqPCR). Shin et al. demonstrated the feasibility of this method by using an allele-specific PCR test (The Cobas *EGFR* Mutation Test from Roche), designed for the detection of 41 hotspot mutations in exons 18–21 [40]. In this study, all analyzed samples showed valid real-time PCR results from both PF supernatants and pellets DNA. Interestingly, the concordance rate between RTqPCR and Sanger sequencing plus PNA-clamping was 98.7%. Another study showed the efficacy of the detection of *EGFR* mutations by RTqPCR in serum and PF of 88 patients with advanced NSCLC cytologically-proven [41]. Interestingly, tumors carrying mutant *EGFR* showed higher response rate to gefitinib compared with *EGFR* wild-type counterparts (90.9 versus 9.1%, *p* < 0.01). An additional work reported successful determination of *EGFR* mutations (exons 18–21) by RTqPCR and direct sequencing in 136 lung adenocarcinoma patients [42]. The authors observed a slight discrepancy in mutation rate between PF and resected tissue. PE harbored higher *EGFR* mutation rate than tissues (68.4 versus 50.5%, *p* = 0.007), which was especially relevant for the L858R mutation (36.8% versus 20.9%, *p* = 0.011). Notably, survival analysis revealed that the median overall survival was longer for TKI-treated patients with *EGFR* mutations than for patients with wild-type *EGFR* (21.4 months versus 11.5 months, *p* = 0.005).

Along with the development of multiplexed detection methods, new works reported promising results with the combination of pyrosequencing, capillary electrophoresis, and RTqPCR [43]. For instance, Akamatsu et al. found genetic abnormalities in 48% of the cytologically-positive PF samples analyzed, and 18% of the cytologically-negative ones. In addition, the concordance rate regarding the mutations found between PE and matched FFPE samples was 88%. Similarly, multiplex RTqPCR detected the oncogenic fusions *KIF5B*–*RET* and *CCDC6*–*RET* in 2.4% of MPE specimens derived from metastatic lung adenocarcinoma patients [31]. Another work showed that the combination of several techniques, such as fluorescence in situ hybridization (FISH) for detecting *ALK* rearrangements, Sanger sequencing for *EGFR*, pyrosequencing for *KRAS* and *BRAF*, and next generation sequencing (NGS) using a lung cancer panel was capable to detect genetic abnormalities in 59% of the PF cell blocks from cytologically-proven NSCLC. Remarkably, authors demonstrated that PF volume was not associated with overall cellularity or tumor cellularity and, therefore, low volume of PF should not discourage further molecular testing [44].

Droplet digital PCR (ddPCR) is a highly sensitive method that allows the detection of very low amounts of DNA, which is the typical scenario for liquid biopsies. PF has been confirmed to be a good specimen for the detection of *EGFR* mutations by ddPCR [45]. In one paper, the mutation detection rate was significantly higher using ddPCR compared with direct DNA sequencing (75.4% versus 43.8%, *p* < 0.0001) and amplification-refractory mutation system (61.3% versus 38.7%, *p* = 0.016) and, more importantly, some patients were redefined for their *EGFR* status. Other authors demonstrated that ddPCR provides high concordance rates of *EGFR* mutations between PF supernatants and cell pellets of lung cancer patients [46].

Buttitta et al. reported the superiority of NGS over Sanger sequencing to assess *EGFR* status in bronchoalveolar lavage and PF from 830 lung cancer patients [47]. For instance, *EGFR* mutations were detectable in 81 and 16% of the cytologically-positive cases for NGS and Sanger sequencing, respectively. Interestingly, while cytological examinations and Sanger sequencing dictated negativity for *EGFR* mutations, 42% of cases turned to be positive according to NGS. Another work compared the performance of NGS with RTqPCR on the detection of *EGFR*, *PIK3CA*, and *KRAS* mutations. Although the concordance was high, NGS was more advantageous than RTqPCR in detecting non-hotspot mutations and providing accurate information about allele sequence and mutation frequency in PF pellets of lung adenocarcinoma patients [48]. Despite good results in PF cell pellets, other studies suggest that NGS on body fluid supernatants possesses higher performance for detecting *EGFR* mutations in cfDNA compared to body fluid sedimentary tumor cells and plasma cfDNA specimens [49]. In addition, targeted hybrid capture deep sequencing for the analysis of 130 cancer-related genes confirmed the presence of mutations in 15 PF supernatants where median depth of target was >1000× [50]. High concordance was seen between PF and FFPE samples.

Later, NGS was shown to successfully identify multiple mutations in 17 genes in 108 PF cell pellets from lung cancer patients [51]. The authors claimed that their NGS approach was currently the most successful allowing a sequencing depth of target of nearly 3492X, 10 times greater than others previously published [52]. Due to the high depth, *EGFR* mutations were detected in 86% of the MPE analyzed, and 66.7% of which were associated with sensitivity to known therapeutic agents. The different genomic approaches utilized in liquid biopsy of PF are summarized in Table 1.

The limit of detection or the sensitivity of the different aforementioned techniques is the main factor that determines the method of choice. Whilst Sanger sequencing and pyrosequencing are capable of providing high versatility on the detection of different genomic alterations, they have the lowest sensitivity [53]. Methods with intermediate sensitivity include high-resolution melting, PNA clamping, RTqPCR, and amplification-refractory mutation system [54], which are especially useful for the detection of single nucleotide variations and small insertions and deletions. Finally, ddPCR and NGS offer the highest sensitivity among all available technologies [55,56]. Limit of detection of different approaches are summarized in Table 2.

##### Approved Clinical Tests for Routine Testing

Although the utility of PF as a source of detectable genomic aberrations seems to be unquestionable, the implementation of the aforementioned techniques in the clinical routine testing is still limited. Recommendations on the development and implementation of safe and effective genetic tests have been suggested by the Evaluation of Genomic Applications in Practice and Prevention (EGAPP). Analytical validity, clinical validity, and clinical utility are the three major criteria taken into account for the adoption of tumor biomarker tests in clinical care [57]. Nevertheless, many commercially available tests have never been submitted to the US Food and Drug Administration (FDA) for approval and, even more, FDA-approved tests have not necessarily needed to demonstrate clinical utility [58].

To date, there are only 3 FDA-approved tests with clinical utility in NSCLC patients, namely Cobas^®^ *EGFR* Mutation Test v2 (Roche Molecular Systems, Inc., Pleasanton, CA, USA), Guardant360^®^ CDx test (Guardant Health, Inc., Redwood City, CA, USA), and The FoundationOne Liquid^®^ CDx test (Foundation Medicine, Inc., Cambridge, MA, USA).

The clinical utility of the Cobas^®^ *EGFR* Mutation Test v2 (Roche Molecular Systems, Inc.) has clearly been demonstrated [59]. This test is a real-time PCR-based method that allows for qualitative identification of 42 *EGFR* mutations in exons 19–21. This test granted FDA approval on 1 June 2016, being the first test employing liquid biopsy for the detection of *EGFR* exon 19 deletions or exon 21 (L858R) mutations in patients with metastatic NSCLC eligible for treatment with erlotinib. This was based on the results from the ENSURE study where patients who carried *EGFR* mutations in plasma were treated with erlotinib presented a higher progression-free survival (PFS) compared to those treated with chemotherapy [60]. The second test, Guardant360^®^ CDx test (Guardant Health, Inc.), is the first test deploying NGS technology in liquid biopsy. It was approved by the FDA on 11 August 2020 for two purposes: (i) as a comprehensive genomic profiling in patients with any solid malignancy, and (ii) as a companion diagnostic to identify NSCLC patients with *EGFR* alterations who may benefit from treatment with osimertinib, based on the evidence of two phase II clinical trials: AURA3 [61] and FLAURA [62,63]. This test consists of the analysis of a 73-gene panel and possesses an average ctDNA detection rate across cancer types of 86%. The third test used in liquid biopsy, the NGS-based FoundationOne^®^ Liquid CDx test (Foundation Medicine, Inc.), was approved by the FDA on 6 November 2020 as a companion diagnostic tool for the detection of genetic aberration in 324 genes by hybridization-based capture technology using ctDNA isolated from plasma specimens. One of the diagnostic indications is the identification of *ALK* rearrangements in patients with NSCLC eligible for treatment with alectinib.

##### Limitations of the Detection of Mutations on ctDNA

The major challenge of liquid biopsy is the detection of infrequent mutation variants, which are typically very close to the limits of sensitivity of the available methodologies. Thus, the quantity of initial input material is crucial to ensure proper detection of all variants. However, ctDNA amount in plasma is low [22], which represents a major concern. Typically, 1 mL of plasma from a cancer patient contains 1500 diploid genome equivalents (10 ng of DNA) with a total quantity of molecules per region of interest of 3000 [64]. This means a theoretical limit of sensitivity of 0.03%, implying the detection of at least one molecule of our target of interest in 3000. If the frequency of mutations is lower, for example 0.01%, we will need more material, which can be sometimes difficult despite the accessibility of blood. So far, the most sensitive techniques, such as CAPP-Seq deep sequencing, are offering sensitivities around 0.02% [65]. In the case of PF, the extraction of several tens of milliliters of PF can be generally guaranteed, which represents a clear advantage over other matrices.

The estimation of tumor burden has received considerable attention as a potential predictive biomarker for treatment response with immune checkpoint inhibitors. However, studies addressing tumor burden through the use of cfDNA from PF are scarce [66].

#### 2.1.2. RNA Species

Although over 75% of the human genome is transcribed, only about 2% are protein-coding genes. The other 98% of transcripts are non-coding RNA (ncRNA), which are grouped into two main categories: housekeeping ncRNAs (such as transfer RNAs and ribosomal RNAs, among others) and regulatory ncRNAs, which can be separated into two groups based on size: small ncRNAs, to which microRNA (miRNA) belong, participating on gene expression regulation; and long non-coding RNAs (lncRNAs), which comprise 82% of the total ncRNAs and participate on tissue homeostasis and cell-type determination [67]. Figure 2 shows a schematic representation of known RNA species.

##### MicroRNA

The presence of miRNAs in blood is widely documented and alterations in their expression have been observed in various solid tumors [68,69], including lung cancer [70,71,72]. Moreover, recent studies have shown that miRNAs are also present in PE, and their characterization could be considered as a diagnostic tool to discriminate between benign and MPE [73,74,75]. Bao et al. identified miRNAs that were differentially expressed in MPE and tuberculous effusions, and three of them became potential diagnostic biomarkers. Furthermore, certain miRNA signatures have been associated with different types of primary lung tumors. For example, the expression of *miR-134*, *miR-185*, and *miR-22* in PF has been linked to NSCLC [76], as well as miR-198 [77]. In another set of analysis, Wang et al. correlated the differential expression of several miRNAs with poor survival rates in NSCLC [78].

##### Long Non-Coding RNA

Regarding lncRNAs, mounting evidence indicates that they are also promising diagnostic and prognostic biomarkers for lung cancer, especially when detected in body fluids. The diagnostic and prognostic potentials of lncRNAs in different lung cancer types have been explored [79]. For instance, many studies showed *MALAT1* to be upregulated in lung tumor tissues compared with the normal ones, suggesting its usefulness as biomarker in different body fluids for early detection of lung cancer [80]. In addition, Wang et al. observed that lncRNAs can also be detected in PF and can be potentially used as tumor markers [81]. In particular, they showed that three lncRNAs (*MALAT1*, *H19*, and *CUDR*) have potential as diagnostic markers in lung cancer-associated MPE. Moreover, baseline *MALAT1* expression in PEs was inversely correlated with chemotherapy response, and the combination of *MALAT1* and carcinoembryonic antigen was even more reliable for diagnostic purposes.

##### Limitations of the ncRNA Implementation

Despite the preceding studies, the use of ncRNAs as diagnostic biomarkers may have certain limitations. For example, the main techniques to detect ncRNA are qRT-PCR-based, and although straightforward and robust, there is still a lack of agreement in the normalization approach. This means that it is difficult to find a suitable internal control to standardize the results. Furthermore, a standardized protocol is needed to guarantee the reproducibility of results on different biological samples (blood and PF). In addition, each ncRNA could regulate more than one messenger RNA (mRNA) or gene, making the determination of a target molecule difficult; while on the other hand, one specific mRNA may be controlled by several miRNAs and lncRNAs; not to mention the implication of the same marker in entirely different diseases [75]. Finally, successful RNA detection is strictly dependent on proper sample handling, where the action of activated RNAses could degrade the substrate, hampering further analysis [82,83].

### 2.2. Exosomes

Extracellular vesicles (EVs), also known as exosomes, are bilayer lipid vesicles of 30–150 nm in size that participate in cell-to-cell communication (e.g., between the stroma and cancer cells) [84]. EVs can contain DNA, mRNA, miRNA, lncRNA, receptors, transcription factors, enzymes, lipids, and extracellular matrix proteins, which serve as communication molecules [84]. It is hypothesized that EVs packaged with those molecules intervene in crucial tumor progression processes such as extracellular matrix remodeling, hypoxia, epithelial to mesenchymal transition, immunosuppression, and vascular leakiness [84]. Interestingly, tumor-derived EVs isolated from serum recapitulate the genetic aberrations present in primary tumors, as the presence of the mutant *EGFR* and *EGFR*vIII in glioblastoma [85] or *MYC* amplification in medulloblastoma [86]. Moreover, circulating tumor-derived EVs have been considered to be promising biomarkers for cancer diagnostic and prognosis. For example, exosomal *miR-21* has been associated with esophageal cancer recurrence and distant metastasis [87].

Andre et al. carried out the first characterization of exosomes in MPE. They were 80 nm in size, rich in antigen-presenting molecules and tumor antigens, and the ones derived from melanoma contained Mart1, which is necessary for the presentation to cytotoxic T lymphocytes [88]. Concerning other immunological properties, Wada et al. found that MPE-derived exosomes bound to TGF-β increase regulatory T-cell number, which could lead to tumor immune evasion [89]. Alegre et al. detected the presence of ubiquitinated human leukocyte antigen-G (HLA-G) in exosomes from pleural exudates, the function of which is to be elucidated [90]. Exosomes from MPEs have also been characterized proteomically [91,92]. Regarding miRNA content, one study found that high expression of *miR-21*, *miR-23b*, and *miR-29* in exosomes from MPEs of ovarian carcinoma were associated with poor progression-free survival, whereas high *miR-21* was correlated with poor overall survival [93]. Mice treated with these exosomes harbored more aggressive tumors [93]. The same exosomal *miR-21* was found in PFs and it identified malignancy better than cytology [94].

The authors pointed out the value of EV-*miR-21* as a diagnostic and prognostic factor of pleural invasions. Moreover, they suggested the involvement of EV-*miR-21* in the mesothelial-to-mesenchymal transition, leading to the phenotypic switch of mesothelial cells to cancer-associated fibroblasts [94]. Other exosomal miRNAs, *miR-205-5p* and *miR-200b*, were increased in lung cancer PEs compared to pneumonia and tuberculosis-related PEs, indicating that these two differentially expressed miRNAs could possess diagnostic value [95]. Similarly, exosomal *miR-182* and *miR-210* were found to be significantly more expressed in MPE from lung adenocarcinoma patients compared to benign PEs [96]. A more extensive RNA profiling study of exosomes derived from PEs identified differential expression of 17 miRNAs and 17 mRNAs between patients with lung adenocarcinoma and benign inflammatory processes. In particular, the *miR-200* family and the mRNA transcript lipocalin-2 presented the strongest diagnostic power [97]. Another study identified the EV-associated miRNAs *miRNA-1-3p*, *miRNA-144-5p*, and *miRNA-150-5p* in PEs through interrogation of 754 miRNAs and Machine Learning as the most accurate lung cancer diagnostic biomarkers [98]. Regarding DNA species, Song et al. studied the feasibility of using exosomal DNA from PEs due to lung adenocarcinoma patients for genetic testing. The study revealed that 78% of the mutations found in PE-derived exosomal DNA matched with those found in PE-derived ctDNA [99], supporting its reliability for genetic testing.

Undoubtedly, EV-miRNAs are informative biomarkers in liquid biopsies with a proven valuable diagnostic and prognostic potential. Some limitations that EV have to overcome are the technicalities related to their isolation, which could hamper implementation in the routine clinical testing. Isolation involves ultracentrifugation and some hospital services may not have access to it. Briefly, pleural exudates are centrifuged a few times at increasing centrifuge forces, usually from 300× *g* to 10,000× *g* for 30 min, to discard dead cells and cell debris. Then, supernatants are added in a 30% sucrose/D2O cushion and ultracentrifuged two times at around 100,000× *g* for at least 1 h. The resulting pellets are the exosomes which can be resuspended in PBS and stored at −80 °C. Exosomes can be quantified for protein content following, for example, the Bradford method. Also, researchers can proceed to the lysis using protein and RNA specific reagents for protein and RNA extraction, respectively.

Apart from being cancer biomarkers, one interesting applicability of exosomes is their use as delivery vehicles for bioactive compounds such as small molecules, siRNA, and cDNA [100]. They are less immunogenic than nanoparticles, could potentially improve the pharmacokinetic properties of nanoparticles, and also deliver gene therapy, viruses, and antibodies inside the specific target cell more efficiently [100].

### 2.3. Cell Entities

PEs contain diverse cell types in different proportions depending on whether they are benign or malignant [101]. Unfortunately, the sensitivity of the cytology in the diagnosis of a MPE is low, around 60%, mostly because nuclei atypia can be confounded with reactive mesothelial cells even for experienced cytopathologists [102]. The detection of certain cell types in the PF, such as CTCs, can be a promising tool that can serve for MPE diagnosis and, at the same time, inform about cancer progression and prognosis. Nevertheless, the isolation of CTCs can be tedious. In this section, we will review the clinical value of CTCs together with the current stage of identification and characterization techniques, and their limitations and challenges.

### 2.4. Circulating Tumor Cells

CTCs are defined as carcinoma cells that have shed into the blood or lymphatic circulation from an existing primary tumor or metastatic lesion. It is estimated that growing tumors shed 3.2 × 10^6^ cells daily per gram of tissue into efferent blood [103]. CTCs are believed to generate from specific tumor subclones that have acquired survival advantage during dissemination and have been able to escape immune surveillance. Altogether, this makes them prone to “seed” in other organs and originate metastasis [104]. Interestingly, Liu et al. reported that the CTCs present in the 4T1 breast cancer model retained the epithelial pheno-type less than the disseminated tumor cells found in the bone marrow [105]. The first accurate characterization of CTCs was reported in 1998 by Racila et al. In this work, the authors detected and isolated CTCs in the blood of patients with breast and prostate cancer using magnetic ferrofluids coupled with an epithelial cell adhesion molecule (EpCAM) antibody followed by cell cytometry [106]. Since then, new technical approaches have emerged for the detection of CTC in PE (Table 3). A general review on the approaches of isolation and characterization of CTC can be found elsewhere [107].

CTCs have been considered to have a clear clinical diagnostic and prognostic value in several malignancies, such as metastatic breast cancer [118], NSCLC [119], colorectal cancer [120], and prostate cancer [121]. Apart from blood, CTCs can be found in rarer human fluids such as PF, cerebrospinal fluid, or urine. An accumulation of PF within the pleural space can be the consequence of cancer progression, usually NSCLC, breast cancer, Kaposi sarcoma, and lymphoma [102]. The detection of malignant cells in the PE denotes advanced stage of the disease usually coursing with metastasis [122]. It also indicates poor prognosis and uselessness of curative treatment. Therefore, the identification of CTCs in the PF represents a valuable approach for the diagnosis of malignancy and can provide guidance for treatment decision. Whilst the detection and characterization of CTCs and the significance of their abundance in peripheral blood have extensively been studied, only few studies have addressed these points in the PF.

One study found that CTCs in PF matched with CTCs in blood from the same patients with NSCLC, regarding cytokeratin expression and absence of EpCAM [108]. The CTC identification was done in two steps: a first CD45 depletion to discard the leukocytes and a subsequent immunostaining with anti-cytokeratin and anti-EpCAM. In another work, the authors established a CTC line from a MPE (SCLC26A) and another from the peripheral blood of a SCLC patient. They reported expression of chitinase-3-like-1/YKL-40 (CHI3L1), a secreted glycoprotein present in cancers with poor outcome, in SCLC26A cells, which correlated with higher expression of several tyrosine kinase receptors responsible for motility, invasion, and poor prognosis [109]. The CTC line SCLC26A also showed to be more sensitive to the combined treatment with camptothecin analogues compared to the SCLC cell lines NCI-H417 and DMS153 [110]. Moreover, SCLC26A expressed E-cadherin at high levels and presented lower expression of stem cell markers compared to other blood CTCs from primary tumors and metastases [111]. In addition, Ge et al. attempted the characterization of a CTC line isolated by anti-EpCAM immunocapture from a MPE secondary to pancreatic cancer. The study revealed polyploidy according to FISH of chromosome 8 [114]. A more complex study was performed by Tang et al. in which they detected CTCs from a MPE by high-throughput fluorescent screening using a fluorescent glucose analogue based on the fact that cancer cells show higher glucose consumption. The validation of malignancy was done by single-cell sequencing [115]. In another study, researchers were able to discriminate between CTCs and other cell types from MPEs, such as leukocytes, with the utilization of a live-cell fluorescent dye, which was able to bind to mitochondria, considering the increased number of mitochondria and higher activity of cancer cells [116]. Such procedure of CTC detection did not disrupt downstream single-cell analysis.

The preceding identification methodologies are mainly indicated for the detection of big CTCs or EpCAM^+^/CK^+^. However, tumors are highly heterogenous and so can be the CTCs, namely highly heterogenous rare CTCs (CRCs) that include circulating endothelial (CECs) and CTCs. Lin et al. proposed an integrative program for the isolation and enrichment of CRCs, which includes subtraction enrichment immunostaining-fluorescence in situ hybridization (SE-iFISH), without hypotonic damage nor anti-EpCAM detection, to enable proper karyotyping for the observation of chromosomal aneuploidy, detection of multiple protein expression and cytogenetic arrangements. Altogether, it can be very informative of the CTC-CEC crosstalk in relation to tumor development processes, such as tumor angiogenesis, drug resistance, and metastasis [123]. The same authors identified individual aneuploid CD31^+^ cells as well as fusion clusters of endothelial-epithelial aneuploid tumor cells with SE-iFISH technology in several biofluids including PE [117].

Despite all these promising techniques, CELLSEARCH^®^ still remains the only clinically approved technology to detect and enumerate CTCs of epithelial origin. It was approved in 2014 by the FDA for the detection of CTCs in peripheral blood and prediction of clinical outcome in patients with metastatic breast cancer [124]. It consists of magnetic nanoparticles coating an anti-EpCAM antibody for the capture and further immunolabelling with cytokeratin 8, 18, and 19 to identify epithelial-differentiating CTCs, with CD45 to discard leucocytes and DAPI to detect nucleated cells. The potential of CELLSEARCH^®^ for the diagnosis of MPE has been explored and compared with traditional cytology [112]. However, its clinical suitability is not clear and more studies are required. A variation of CELLSEARCH^®^ was used by Beije et al. for the identification and enumeration of CTCs in PFs of malignant mesothelioma patients. The methodological variation includes the immunodetection by flow cytometry of the melanoma cell adhesion molecule (MCAM), which is specific for malignant cells [125].

Studies including analysis of exosomes and CTC in PF are scarce. This explains why it is difficult to refute results from different reports or make a contrasted review about these two biomarkers.

## 3. Future Perspectives: Tumor-Educated Cells

Apart from ctDNA, RNA species, and CTCs, there are other cellular entities present in the blood that have started receiving interest as diagnostic and prognostic cancer biomarkers. Eventually, they will start getting studied in MPE. These are tumor-associated cells such as tumor-educated platelets (TEPs) and tumor-associated macrophages (TAM). Platelets emerge from the fragmentation of megakaryocytes in the bone marrow and are anucleated. Primarily known to be involved in wound healing processes, their newly discovered role in tumor dissemination has awaken the interest of researchers. Tumors are in crosstalk with platelets and the exchange and sequestration of tumor-associated molecules by platelets contribute to their “education” [126]. Platelets have the capacity to undergo splicing of mRNAs when activated via surface receptors and lipopolysaccharides, or by signals emitted by cancer cells and the tumor microenvironment [127]. Interestingly, the different abundances of mRNA, as a result of platelets activity, can serve as a cancer diagnostic tool [128]. For example, a study revealed a platelet-based gene expression signature of metastatic lung cancer characterized by a decreased expression of 200 genes, with the most altered levels between metastatic and healthy subjects [129]. Another study has proposed the analysis of the TEP mRNA repertoire in blood as a promising early pan-cancer diagnosis tool, based on the results obtained with RNA Sequencing in TEPs [130]. Notably, TEP RNA has been considered to be a more advantageous biomarker for the detection of early-stage cancers compared to the ctDNA in plasma due to the low abundance of the latter [128]. Despite the high accuracy of TEP RNA to differentiate between malignant and benign nature of diseases, there are still no works in the literature studying their potential to discriminate between benign and MPE, which we believe is nonetheless worth exploring.

Another understudied field in MPEs is the identification of TAMs. TAMs are the most abundant cell type of the tumor microenvironment. TAMs, like CTCs, derive from the primary tumor and are believed to support CTCs dissemination to distant sites [131]. For that, cell-to-cell communication is needed via production of chemoattractants by TAMs, including MMP-1 and CXCL12 [131]. It is hypothesized that the presence of TAMs in plasma is a biomarker of advanced cancer. In this line, Adams et al. isolated TAMs by microfiltration from peripheral blood of patients with breast, pancreatic and prostate cancer, but these cells were not present in healthy individuals [132]. TAMs expressed epithelial, monocytic and endothelial markers and were bound to CTCs. In another study, it was shown that the lncRNA *LINC00662* could enhanced hepatocellular growth and metastasis through Wnt/β-catenin signaling and M2 TAM polarization [133]. Like TEPs, little is known about the potential of TAMs as biomarkers of MPEs, though they could provide valuable diagnostic and predictive information.

Among all strategies evaluated in this review, the choice of one technique over the others will depend on factors such as the sample quality and quantity, prior molecular knowledge on specific tumor gene characteristics, available budget, and equipment requirements. For example, PCR-based approaches, including direct PCR sequencing, PNA-PCR, RTqPCR, and ddPCR, offer a reasonable performance; sensitivity being the main parameter that differentiates them. They are in general affordable but require some previous molecular biology background. Other technologies like NGS-based approaches are quite expensive, but they allow the detection of multiple mutations and less sample volume is generally necessary. In addition, prior knowledge is not needed, though bioinformatics skills are essential to perform the variant calling. Regarding RNA analysis, sample quality is the major concern. RNA detection is quite simple and affordable; however, it requires some specialized training. Exosome studies are very innovative and, in our opinion, will be crucial for further advances in the field. However, exosome extraction is tedious, and the identification of their cargo relies on extremely sensitive techniques. Finally, cell-based research is technically quite complex because of the low abundance of CTC. Moreover, it is imperative to reproduce physiological conditions to obtain good results, a specialized equipment is mandatory, and test reproducibility is a major challenge.

## 4. Conclusions

PF provides a valuable opportunity for cancer detection, which is especially useful in those cases where tissue biopsy is not available. In this regard, genomic analysis of PFs has a positive impact on diagnostic performance and faithfully recapitulates the molecular profiles found in tissue. It is worth noting that PF is a very suitable matrix for the detection of actionable mutations of the most studied oncogenes and can be used as a guide for therapeutic decisions. Interestingly, cell-free specimens such as plasma and PF supernatants have revolutionized the way we understand the genomic landscape of tumors. Indeed, they are the samples of choice for state-of-art genomic characterization approaches as they ensure sample quality and consistency, unlike other matrices like tissue, still considered as the gold standard. Nevertheless, there are yet major concerns to be addressed related to the implementation of the liquid biopsy in the clinical practice. In the case of PFs, it is still to be considered as a valid sample.

Unlike DNA molecules, RNA species are far away from the routine clinical practice; however, they are promising and detectable on PE specimens. Regarding CTCs, their detection opens an avenue of new opportunities to understand tumor biology processes and infer clinical interventions in the future. Although not yet detected in PFs, the exploration of educated cells represents a new paradigm, which could provide valuable information added to the global tumoral picture and improve our understanding about the dynamic networks existing between tumor and stroma. Undoubtedly, PF has begun to stand out above other matrices and will be a key biological sample to understand MPE and cancers coursing with MPE.

## Figures and Tables

**Figure 1 cancers-13-02798-f001:**
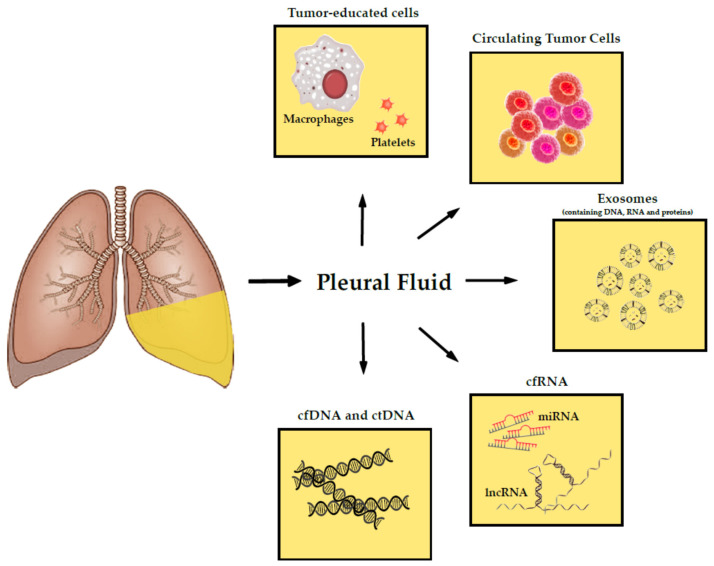
Schematic representation of the foremost tumor-derived products in pleural fluid. Abbreviations: deoxyribonucleic acid (DNA), cell-free DNA (cfDNA), cell-free RNA (cfRNA), circulating tumor DNA (ctDNA), ribonucleic acid (RNA), microRNA (miRNA), long non-coding RNA (lncRNA).

**Figure 2 cancers-13-02798-f002:**
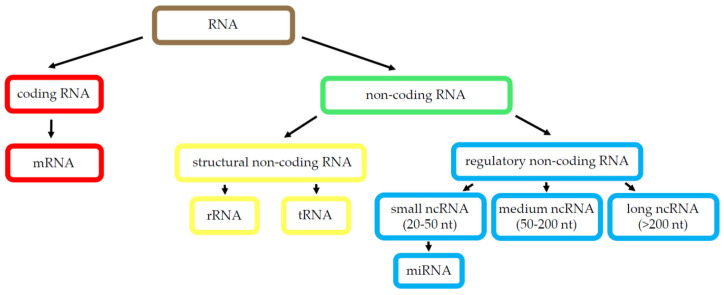
RNA species. Scheme of the best-known RNA types. Abbreviations: ribonucleic acid (RNA), messenger RNA (mRNA), non-coding RNA (ncRNA), ribosomal RNA (rRNA), transfer RNA (tRNA), microRNA (miRNA), long non-coding RNA (lncRNA).

**Table 1 cancers-13-02798-t001:** Genomic approaches of liquid biopsies on pleural fluid specimens.

Ref	Year	Genomic Approach	Patients and Samples	Mutations Detected	Detection Rate	Concordance	Relevance
[29]	2005	PCR and direct DNA sequencing	1 NSCLC patient: 1 PF	*EGFR*: exon 19 delE746-A750	100%	No *EGFR* status tested on tissue	Responsiveness to gefitinib therapy
[30]	2014	HRM and direct DNA sequencing	36 NSCLC patients: 36 PFs and cell blocks, some of them paired with 22 tumor tissues	*EGFR*: exon 19 del; exon 21 L858R and exon 20 Ins773	50% in PFs 36.1% in cell blocks 59.1% in tissue	90.8% PF-tissue 72.2% PF-cell block	PF cell-free supernatant is a better source for mutation detection than cell blocks
[31]	2015	Multiplex RTqPCR and DNA sequencing	722 lung adenocarcinoma patients: 722 PF cell blocks	*RET*: KIF5B-*RET* and CCDC6-*RET*; *EGFR*: exon 19 del, exon 20, exon 21 L858R; *KRAS*: codons 12, 13, and 61; *ALK*: *EML4*-*ALK*	2.4% *RET* 62.5% *EGFR* 1.8% *KRAS* 7.1% *ALK*	Not evaluated	Metastatic *RET*-rearranged lung adenocarcinoma patients have better prognosis than metastatic *EGFR*-mutant tumors
[32]	2011	Two-round PCR and DNA sequencing	37 lung adenocarcinoma patients: 37 paired primary and metastatic tumors; 21 paired primary tumor and pleural metastases (PF cell blocks or pleural biopsy)	*EGFR*: exon 19 del; exon 21 L858R *KRAS*: codon 12–13	*EGFR*: 48.6% in primary tumor; 43.2% in metastatic tumor and 38.1% in pleural metastasis *KRAS*: 2.7% in primary tumor and 5.4% in metastases	*EGFR*: 83.8% primary tumor-metastatic tumor; 85.7% primary tumor-pleural metastases *KRAS*: 50% primary tumor-metastatic tumor	Responsiveness to *EGFR* TKI is more likely to be correlated with *EGFR* status in metastatic lesions than in primary tumors
[33]	2008	PCR and PCR-RFLP followed by direct DNA sequencing	26 NSCLC patients: 26 PFs paired with PF cell blocks	*EGFR*: exon 21 L858R; exon 19 delE746-A750 and E747–749delA750P; exon 20 S768I	PCR: 23.1% in both cell blocks and PFs PCR-RFLP: 50% in both cell blocks and PFs	PCR: 71.4% PF-cell blocks PCR-RFLP: 100% PF-cell blocks	PF cell-free supernatants and cell blocks are feasible clinical specimens for *EGFR* mutation detection
[34]	2006	PCR and direct DNA sequencing	43 NSCLC patients: 43 PFs	*EGFR*: exon 19 E746_A750del, E746_T751del insA, and L747_T751del; exon 21 L858R	25.6% in PFs	Not evaluated	*EGFR* status in PF is a predictor of response to gefitinib
[35]	2017	Microarray hybridation and NGS	11 cancer patients with different primary tumors: lung, breast, pancreas, colon, rectal, and renal. Samples included ascites, PFs and plasma	Mutations on 9 gene-panel: *BRAF*, *EGFR*, *IDH1*, *IDH2*, *KRAS*, *NRAS*, *PIK3CA*, *PTEN*, and *TP53*	63% of patients had proper material to be analyzed 1 patient with CNV’s mutation gain in PF	Not evaluated	cfDNA from ascites and PFs provide additional information not detected in tumor and/or plasma
[36]	2018	PNA clamping	50 lung adenocarcinoma patients: PF and EV PF	*EGFR*: exon 18 G719S, exon 19 del, exon 21 L858R	59.4% in tissue 68.8% in EV PF 59.4% in PFs	88% tissue-PF 91% tissue-EV PF 91% PF-EV PF	Liquid biopsy using EV-derived DNA is promising for *EGFR* genotyping, including the detection of the T790M resistance mutation
[37]	2018	PNA clamping	40 lung adenocarcinoma patients: 40 PF cell blocks paired with 23 primary tissues	*EGFR*: exon 19 del, exon 21 L858R, exon 21 L861R	72.5% in PFs 82.6% in tissue	73.9% tissue-PF	*EGFR* mutation status in PF cell block is highly predictive of *EGFR* TKI efficacy
[38]	2013	PNA clamping and direct DNA sequencing	37 NSCLC patients: tumor tissue, PF, PF cell blocks, and serum	*EGFR*: exon 18 G719X, exon 19 del, exon 20 S768I and L788L, exon 21 L858R, L861Q, and R832H	PNA: 35.7% in tissue, 33.3% in PF cell blocks, 27% in PFs, and 2.8% in serum Seq: 27.8% in tissue, 38.1% in PF cell blocks, 27% in PFs, and 2.8% in serum	86% PNA-Seq in tissue 95% PNA-Seq in PF cell blocks 89% PNA-Seq in PFs 94% PNA-Seq in serum	PF has good diagnostic performance and PNA clamping method offers sensitive and accurate detection of *EGFR* allowing a better prediction of response to *EGFR* TKI
[39]	2012	PNA clamping and direct DNA sequencing	NSCLC patients: 41 PF cell blocks and 23 lung biopsies and/or resected tissues	*EGFR*: exon 19 del, exon 21 L858R, and L861Q	PNA: 39% in PF cell blocks and 69.6% in tissues Seq: 14.6% in PF cell blocks and 52.2 in tissues	PNA: 82.6% biopsy-resected tissues Seq: 100% biopsy-resected tissues	Both PNA clamping and DNA sequencing methods are complementary for the analysis of *EGFR* status in lung adenocarcinoma patients
[40]	2017	RTqPCR, PNA clamping, and DNA sequencing	77 NSCLC patients: 77 PFs and cell blocks	*EGFR*: exon 18 G719X, exon 19 del, exon 20 T790M E20 insertion, exon 21 L858R	RTqPCR: 45.5% in PFs and/or PF cell blocks	98.7% RTqPCR-PNA DNA seq	RTqPCR on follow-up PF samples is a promising approach when tissue is difficult to obtain
[41]	2010	RTqPCR	88 NSCLC patients: 32 PFs and 56 plasma specimens	*EGFR*: exon 19 E746_A750del and L747–S752del, exon 21 L858R	23.2% in plasma 28.1% in PF	Not evaluated	*EGFR* mutation detection by RTqPCR highly predicted the efficacy of gefitinib in advanced NSCLC
[42]	2008	RTqPCR and direct DNA sequencing	136 lung adenocarcinoma patients: 136 cytologically-positive PF cell blocks; 91 lung adenocarcinoma resected patients: 91 tissue biopsies	*EGFR*: exon 21 L858R, L861Q and K861I; exon 19 del; exon 20 767–769 dupASV, 771_H773insYNP and H773Y	50.5% in tumor tissue 68.4% in PF cell blocks	Not evaluated	Patients with MPE had a higher *EGFR* mutation rate than the surgically resected specimens suggesting *EGFR* TKI treatment for those patients having MPE
[43]	2014	Pyrosequencing and RTqPCR	84 lung cancer patients: 102 PF cell blocks	*EGFR*, *KRAS*, *BRAF*, *PIK3CA*, *NRAS*, *MEK1*, *AKT*, *PTEN*, and *ERBB2*	Global mutation rate: 42%. Per gene: *EGFR*: 29%, *ALK*: 5%, *KRAS*: 4%	88% PF and FFPE tissue	Multiplexed molecular testing is suitable to monitor molecular profiles in PF
[44]	2017	FISH, Sanger seq, pyrosequencing, and NGS	50 NSCLC patients: 27 PF cell blocks suitable for genetic analysis	*EGFR*: exon 21 L858R; exon 19 E746_A750del and L747_A750del, S768I and V769; exon 20 T790M; *KRAS*: codon 12 and 13 *PI3KCA*: E542K *ALK*: *EML4*-*ALK* fusion	Global mutation rate: 59% Per gene: *EGFR*: 33%, *KRAS*: 25.9%, PI3KCA: 3.7% and *ALK*: 3.7%	Not evaluated	Molecular profiling of PF is a viable alternative to testing solid tissue in advanced NSCLC
[45]	2017	Digital droplet PCR, ARMS, and direct PCR sequencing	95 NSCLC patients: PF	*EGFR*: exon 19 A750-K757del and S752-L760del, exon 21 L858R	ddPCR: 75.4% ARMS: 61.3% Sanger seq: 43.8%	Not evaluated	ddPCR is feasible to detect *EGFR* mutations in PF
[46]	2018	Digital droplet PCR	90 cancer patients (74 lung cancer): 80 bronchial lavages, 9 PFs, and 1 cerebrospinal fluid	*EGFR*: exon 19 E746-A750del, exon 20 T790M and exon 21 L858R	13.3% in supernatants 16.7% in cell blocks	96.7% supernatant-cell block	Cell-free supernatants are suitable for genetic analysis using ddPCR
[47]	2013	NGS	830 lung adenocarcinoma patients: 33 bronchial lavages and 15 PF cell blocks	*EGFR*: exon 19del, exon 21	81% by NGS 16% by Sanger seq	53.3% PF cell block-tissue 78.8% bronchial lavage cell block-tissue	NGS is a sensitive method for the detection of mutations in liquid specimens
[48]	2018	NGS, RTqPCR, and Sanger seq	18 lung adenocarcinoma patients: 8 PF cell blocks and 10 tissues	*EGFR*: exon 19 E746-A750del, E746-T751del, S752-I759del, and T790M, exon 21 L858R; *KRAS*: G13C; *PIK3CA*: E545K	46.7% by NGS 40.0% by RTqPCR 33.3% by Sanger seq	75% PF cell block-tissue	High quality of DNA from FFPE PF cell blocks assures successful NGS
[49]	2019	NGS	20 lung adenocarcinoma patients: 15 PFs, 2 pericardial fluids, 2 cerebrospinal fluids, 1 ascites, and 20 plasmas	*EGFR*: exon 19del, exon 21 L858R	Cell-free fluid: 100% PF, pericardial, ascites, and cerebrospinal; Cell block: 100% in PF, 50% in pericardial fluid and cerebrospinal fluid Plasma: 80%	100% PF-PF cell block 50% pericardial fluid-pericardial fluid cell block 50% cerebrospinal fluid-cerebrospinal fluid cell block 86.7% PF-plasma	Cell-free body fluids have a higher detection rate and sensitivity for tumor-specific mutations than body fluid sediments or plasma
[50]	2020	NGS	21 cancer patients (7 lung adenocarcinomas): 15 PFs, 5 peritoneal fluids, 1 pericardial fluid, 8 cell blocks, and 3 tissues	Mutations on 130 gene-panel of cancer related genes	71.4% global mutation rate	72.3% PF-tissue	cfDNA testing of effusion samples allows robust detection of clinically actionable genetic biomarkers
[51]	2020	NGS	108 lung cancer patients: PF cell blocks	Mutations on 17 gene-panel containing lung cancer associated genes	86% *EGFR*, 41.7% *TP53*, 9% BCL2, 21.3% *BRAF*, 19.4% *PIK3CA*, 21.3% *PTEN*, 18.5% FGFR1, 25% *MET*, 27.8% *RET*, 5.6% *KRAS*, and 5.6% *ALK*	Not evaluated	High capture efficiency and deep sequencing using NGS for molecular profiling of pleural effusions
[52]	2018	NGS and ARMS-PCR	30 NSCLC patients: PF cell blocks and tissues	Mutations on 9 gene-panel containing lung cancer associated genes	Global mutation rate: 83.33% in both tissue and PF cell blocks	86.7% PF cell blocks-tissue	High concordance rate on the detection of *EGFR*, *KRAS*, and *ALK* between tissue and PF cell blocks

Abbreviations; ARMS: amplification-refractory mutation system; CNV: copy number variation, EV: extracellular vesicles; FISH: fluorescence in situ hybridization, FFPE: formalin-fixed and paraffin-embedded; HRM: high resolution melting, MPE, malignant pleural effusion; NGS: next generation sequencing; NSCLC: non-small cell lung cancer; PCR: polymerase chain reaction; PF: pleural fluid; PNA: peptide nucleic acid; RFLP: restriction fragment length polymorphism; TKI: tyrosine kinase inhibitor. When comparing PF with plasma or serum, it has been observed that the detection rate for EGFR mutations is higher in the former [38,41,49]. One study demonstrated a full detection rate of EGFR status both in cell-free PF and other cell-free matrices such as pericardial fluid, cerebrospinal fluid, and ascites [49].

**Table 2 cancers-13-02798-t002:** Limit of detection of different genomic approaches to detect mutation variants.

Genomic Approach	Limit of Detection
**Sanger sequencing**	20–25%
**Pyrosequencing**	5–10%
**High resolution melting**	5%
**Peptide nucleic acid clamping**	5%
**Real-time quantitative PCR**	0.5–5%
**Amplification-refractory mutation system**	1%
**ddPCR**	0.001–0.01%
**NGS-Safe seq**	0.1%
**NGS-CAPP seq**	0.01%

Abbreviations: ddPCR: digital droplet PCR; NGS-CAPP seq: next generation sequencing-cancer personalized profiling by deep sequencing.

**Table 3 cancers-13-02798-t003:** Studies exploring the detection and characterization of CTC by different techniques in pleural effusions.

Name of the CTC Line	Primary Tumor	Technique for CTCs Isolation/Enrichment/Detection/Characterization	Major Findings/Basic Characterization	Reference
**Not mentioned**	Non-small cell lung cancer	CD45 depletion, and EpCAM and CD45 immunostaining	CD45^−^ EpCAM^−^ CK^+^ CTCs were identified in PEs with CTCs detected in matched peripheral blood	[108]
**SCLC26A**	Pancreatic cancer	Not mentioned	CTC had no expression of CHI3L1 and lower expression of tyrosine kinase receptors	[109]
**SCLC26A**	Pancreatic cancer	Not mentioned	CTCs were sensitive to camptothecin analogues	[110]
**SCLC26A**	Pancreatic cancer	Not mentioned	High expression of E-cadherin consistent with a mesenchymal-epithelial transition. Low expression of stem cell markers, which hamper the epithelial phenotype	[111]
**Not mentioned**	Benign, malignant epithelial, and non-epithelial cancers	Anti-EpCAM capture and CELLSEARCH^®^	More studies are needed to implement pleural CELLSEARCH. It could be useful in addition to traditional cytology	[112]
**Not mentioned**	Malignant mesothelioma	Variation of CELLSEARCH^®^ for CTC enumeration based on the detection of MCAM	MCAM^+^ cells are malignant. Thus, they can help to identify malignancy in PE	[113]
**Not mentioned**	Pancreatic cancer	Anti-EpCAM capture and iFISH	CTCs were non-hematopoietic (CD45^−^) and polyploid	[114]
**Not mentioned**	Lung cancer	Identification of CTC by using a fluorescent glucose analog (2-NBDG) by high-throughput screening. Confirmation of CTC by single-cell sequencing	Most CTCs share the same oncogenic mutations as the primary tumors where they come from. Detection of emerging secondary mutations responsible for drug resistance before the manifestation of resistance. The reported method can complement traditional cytology for MPE diagnosis	[115]
**Not mentioned**	Lung and liver cancer	Mitochondria-targeting bioprobe with aggregation-induced emission activity. CTC characterization by single-cell sequencing	With this method of detection of CTCs there is less cell disruption; thus downstream single-cell sequencing gets less affected than using traditional cytokeratin markers	[116]
**Not mentioned**	Not mentioned	Antigen-independent subtraction enrichment and immunostaining-FISH (SE-iFISH)	Aneuploid circulating rare cells can be detected using the methodology described	[117]

Abbreviations: CHI3L1: chitinase 3 Like 1; CTC: circulating tumor cells; CK: cytokeratin; EpCAM: epithelial cell adhesion molecule; iFISH: immunostaining fluorescent in situ hybridization; MCAM: melanoma cell adhesion molecule; SE: subtraction enrichment.

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
