# Peer review of "Diving into the Pleural Fluid: Liquid Biopsy for Metastatic Malignant Pleural Effusions"

_cancers, 2021, doi:10.3390/cancers13112798_

Round 1
Reviewer 1 Report
The paper described very well the successful experiments using pleural fluid for liquid biopsy and is very relevant for the community.
In order to improve it even further, I'd suggest a few modifications:
- try to include papers reporting negative results if available, so that the limitations of this strategy are clearer.
- include a comparison between the different biomarkers, as well as the techniques described, indicating for future users what would be the best choices in different circumstances.
- provide some background information on the differences between exudative and transudative PF, and why the content of cfDNA in them is so different.
- provide some background information of why one technique outperform the others, e.g. PNA vs. PCR. Is it something observed for other matrices, or is it exclusive of PF, why?
- provide more details on how PF compare with other matrices, such as plasma. In which circumstances would it be advantageous to use one vs. the other.
Author Response
Answer to Reviewer 1
Comments
The paper described very well the successful experiments using pleural fluid for liquid biopsy and is very relevant for the community.
In order to improve it even further, I'd suggest a few modifications:
Point 1: try to include papers reporting negative results if available, so that the limitations of this strategy are clearer.
Response 1. We would like to thank the Reviewer for pointing this out. Although it’s difficult to find negative results in the literature concerning the use of pleural fluid for searching of biomarkers of malignancy, we have added some references that expose limitations and other aspects that need further investigation for each strategy presented in the review:
Lines 377-379: “The estimation of tumor burden has received considerable attention as a potential predictive biomarker for treatment response with immune checkpoint inhibitors. However, studies addressing tumor burden through the use of cfDNA from PF are scarce [67]”
Lines 399-401: “Finally, successful RNA detection is strictly dependent on proper sample handling, where the action of activated RNAses could degrade the substrate hampering further analysis [83,84]”.
Lines 563-565: “Studies including analysis of exosomes and CTC in PF are scarce. This explains why it is difficult to refute results from different reports or make a contrasted review about these two biomarkers”.
Point 2: include a comparison between the different biomarkers, as well as the techniques described, indicating for future users what would be the best choices in different circumstances.
Response 2. We find this comment very useful. To address this point, we have included a paragraph before the conclusions that provides a critical view of the different biomarkers analyzed to detect malignancy:
Lines 607-623: “Among all strategies evaluated in this review, the choice of one technique over the others will depend on factors such as the sample quality and quantity, prior molecular knowledge on specific tumor gene characteristics, available budget, and equipment requirements. For example, PCR-based approaches, including direct PCR sequencing, PNA-PCR, RTqPCR and ddPCR, offer a reasonable performance; sensitivity being the main parameter that differentiates them. They are in general affordable but require some previous molecular biology background. Other technologies like NGS-based approaches are quite expensive, but they allow the detection of multiple mutations and less sample volume is generally necessary. In addition, prior knowledge is not needed, though bioinformatics skills are essential to perform the variant calling. Regarding RNA analysis, sample quality is the major concern. RNA detection is quite simple and affordable; however, it requires some specialized training. Exosome studies are very innovative and, in our opinion, will be crucial for further advances in the field. However, exosome extraction is tedious, and the identification of their cargo relies on extremely sensitive techniques. Finally, cell-based research is technically quite complex because of the low abundance of CTC. Moreover, it is imperative to reproduce physiological conditions to obtain good results, a specialized equipment is mandatory, and test reproducibility is a major challenge”.
Point 3: provide some background information on the differences between exudative and transudative PF, and why the content of cfDNA in them is so different.
Response 3. We agree with the Reviewer that the paragraph comparing transudative and exudative PE in terms of cfDNA content needs clarification. Consequently, we have added the following text together with additional references:
Lines 96-103: “A transudate is a plasma ultrafiltrate that usually results from a disequilibrium of the Starling forces (hydrostatic and oncotic pressures) in which a healthy pleura acts as a passive membrane [15]. Therefore, the small amounts of cfDNA detected in transudates derive from plasma rather than local production [14]. In contrast, an exudate implies inflammatory and cellular responses into the pleural cavity due to an increased microvessel permeability and/or lymphatic blockage [16]. The infiltration of leukocytes in infectious PE constitutes a source of cfDNA from dying cells [17], whereas the presence of a pleural tumor increases ctDNA concentrations”.
Point 4: provide some background information of why one technique outperform the others, e.g. PNA vs. PCR. Is it something observed for other matrices, or is it exclusive of PF, why?
Response 4. We would like to thank the Reviewer for this suggestion, which will improve the part about genomic strategies for mutation profiling in pleural fluid. One of the main differences between techniques is their particular limit of detection. We have included a table (Table 2) at the end of point 2.1.1.1. and a brief explanation regarding on the subject.
We did not find evidence suggesting that the performance of these techniques on pleural fluid could depend on the matrix used. The limiting factor in all of them is the initial amount and quality of extracted DNA.
Lines 271-279: “The limit of detection or the sensitivity of the different aforementioned techniques is the main factor that determines the method of choice. Whilst Sanger sequencing and pyrosequencing are capable of providing high versatility on the detection of different genomic alterations, they have the lowest sensitivity [54]. Methods with intermediate sensitivity include high resolution melting, PNA clamping, RTqPCR and amplification-refractory mutation system [55], which are especially useful for the detection of single nucleotide variations and small insertions and deletions. Finally, ddPCR and NGS offers the highest sensitivity among all available technologies [56,57]. Limit of detection of different approaches are summarized in table 2”.
Table 2. Limit of detection of different genomic approaches to detect mutation variants
|
Genomic approach |
Limit of detection |
|
Sanger sequencing |
20-25% |
|
Pyrosequencing |
5-10% |
|
High resolution melting |
5% |
|
Peptide nucleic acid clamping |
5% |
|
Real-time quantitative PCR |
0.5-5% |
|
Amplification-refractory mutation system |
1% |
|
ddPCR |
0.001%-0.01% |
|
NGS-Safe seq |
0.1% |
|
NGS-CAPP seq |
0.01% |
Abbreviations: ddPCR: digital droplet PCR; NGS-CAPP seq: next generation sequencing-cancer personalized profiling by deep sequencing
Point 5: provide more details on how PF compare with other matrices, such as plasma. In which circumstances would it be advantageous to use one vs. the other.
Response 5. We have partially addressed this comment in Table 1 with the comparison between plasma and other matrices regarding the mutation detection rate. Nevertheless, we have included a few lines in the manuscript explaining this point in more detail:
Lines 267-270: “When comparing PF with plasma or serum, it has been observed that the detection rate for EGFR mutations is higher in the former [39,42,50]. One study demonstrated a full detection rate of EGFR status both in cell-free PF and other cell-free matrices such as pericardial fluid, cerebrospinal fluid and ascites [50].
New references added in the text:
[15] Chetty, K. Transudative pleural effusions. Clin. Chest Med. 1985, 6, 49–54.
[16] Porcel, J.M.; Light, R.W. Diagnostic approach to pleural effusion in adults. Am. Fam. Physician 2006.
[17] Santotoribio, J.D.; Cabrera-Alarcón, J.L.; Batalha-Caetano, P.; Macher, H.C.; Guerrero, J.M. Pleural fluid cell-free DNA in parapneumonic pleural effusion. Clin. Biochem. 2015, 48, 1003–1005, doi:10.1016/j.clinbiochem.2015.07.096.
[54] Vanni, I.; Tanda, E.T.; Spagnolo, F.; Andreotti, V.; Bruno, W.; Ghiorzo, P. The Current State of Molecular Testing in the BRAF-Mutated Melanoma Landscape. Front. Mol. Biosci. 2020, 7.
[55] Ye, P.; Cai, P.; Xie, J.; Wei, Y. The diagnostic accuracy of digital PCR, ARMS and NGS for detecting KRAS mutation in cell-free DNA of patients with colorectal cancer: A systematic review and meta-analysis. PLoS One 2021, 16,
[56] Chakrabarti, S.; Xie, H.; Urrutia, R.; Mahipal, A. The promise of circulating tumor DNA (ctDNA) in the management of early-stage colon cancer: A critical review. Cancers (Basel). 2020, 12, 1–18.
[57] Postel, M.; Roosen, A.; Laurent-Puig, P.; Taly, V.; Wang-Renault, S.F. Droplet-based digital PCR and next generation sequencing for monitoring circulating tumor DNA: a cancer diagnostic perspective. Expert Rev. Mol. Diagn. 2018, 18, 7–17, doi:10.1080/14737159.2018.1400384.
[67] Newman, A.M.; Lovejoy, A.F.; Klass, D.M.; Kurtz, D.M.; Chabon, J.J.; Scherer, F.; Stehr, H.; Liu, C.L.; Bratman, S. V.; Say, C.; et al. Integrated digital error suppression for improved detection of circulating tumor DNA. Nat. Biotechnol. 2016, 34, 547–555, doi:10.1038/nbt.3520.
[83] Greillier, L.; Roll, P.; Barlesi, F.; Robaglia-Schlupp, A.; Fraticelli, A.; Cau, P.; Astoul, P. Apport des puces à ADN dans le diagnostic étiologique des pleurésies: Étude de faisabilité. Rev. Mal. Respir. 2007, 24, 859–867, doi:10.1016/S0761-8425(07)91388-1.
[84] Michael, C.W.; Davidson, B. Pre-analytical issues in effusion cytology. Pleura and Peritoneum 2016, 1, 45–56, doi:10.1515/pp-2016-0001.

Reviewer 2 Report
The manuscript proposed by Sorolla et al addresses the role of liquid biopsy for metastatic malignant pleural effusions. The manuscript is well organized and interesting to be read. I have some minor revisions that could clarify some aspects of the paper.
For figure 1. please add briefly the content of the exosomes (DNA, mRNA, ncRNAs, proteins). Also please add cfDNA as later in the text you make reference to this type of structures.
Line 90, please use adjacent normal cells instead of "non-tumor cells"
Please add reference 74 to the miRNA chapter.
I would suggest moving the exosomes chapter after or make a direct correlation with ncRNAs one as a lot of information is related to miRNAs and lncRNAs in this chapter.
Author Response
Answer to Reviewer 2
Comments
The manuscript proposed by Sorolla et al addresses the role of liquid biopsy for metastatic malignant pleural effusions. The manuscript is well organized and interesting to be read. I have some minor revisions that could clarify some aspects of the paper.
Point 1. For figure 1. please add briefly the content of the exosomes (DNA, mRNA, ncRNAs, proteins). Also please add cfDNA as later in the text you make reference to this type of structures.
Response 1. Thank you for the recommendations. We added the exosome content as suggested. Moreover, we changed the title of the DNA box for “cfDNA and ctDNA” and now all DNA species are represented. Consequently, we modified the figure legend by adding the new abbreviations:
Figure 1. Schematic representation of the foremost tumor-derived products in pleural fluid. Abbreviations: deoxyribonucleic acid (DNA), cell-free DNA (cfDNA), cell-free RNA (cfRNA), circulating tumor DNA (ctDNA), ribonucleic acid (RNA), microRNA (miRNA), long non-coding RNA (lncRNA).
Point 2. Line 90, please use adjacent normal cells instead of "non-tumor cells"
Response 2. On line 92 we used the term “non-tumor cell” because the article of Souza et al. described how the cfDNA and ctDNA released from tumor cells to the blood promote malignant transformation of healthy cells, both cells adjacent to the tumor and cells in distant organs causing metastasis.
Point 3. Please add reference 74 to the miRNA chapter.
Response 3. The article by Wojczakowski et al. is a good review which contains abundant information about the presence of miRNAs in pleural effusions and their role as biomarkers. Therefore, we have added it, as suggested, under the miRNA chapter (reference No. 76).
Point 4. I would suggest moving the exosomes chapter after or make a direct correlation with ncRNAs one as a lot of information is related to miRNAs and lncRNAs in this chapter.
Response 4. Following your recommendation, we have moved the exosomes (2.2) chapter after the ncRNA one (2.1.2).
Reviewer 3 Report
Sorolla and colleagues well summarized application of liquid biopsy strategies on malignant pleural effusions (MPFs). Concerning MPFs have plenty of tumor cells, it is highly recommended that authors expand their introduction of “Cell Entities”, particularly circulating tumor cells. A good review very well summarized CTC technology (Yu et al., 2011 J Cell Biol 192:373). In order to more appropriately fit the scope of “Cancers”, recent new progress on CTC detection strategies, such as combined immunostaining and aneuploidy examination should be included (Liu et al., 2019 Sci Adv 5:eaav4275; Lin 2018 Diagnostics (Basel) 8:26; Lin et al., 2017 Sci Rep 7:9789).
Author Response
Answer to Reviewer 3
Comments
Sorolla and colleagues well summarized application of liquid biopsy strategies on malignant pleural effusions (MPFs). Concerning MPFs have plenty of tumor cells, it is highly recommended that authors expand their introduction of “Cell Entities”, particularly circulating tumor cells. A good review very well summarized CTC technology (Yu et al., 2011 J Cell Biol 192:373). In order to more appropriately fit the scope of “Cancers”, recent new progress on CTC detection strategies, such as combined immunostaining and aneuploidy examination should be included (Liu et al., 2019 Sci Adv 5:eaav4275; Lin 2018 Diagnostics (Basel) 8:26; Lin et al., 2017 Sci Rep 7:9789).
Answer
We would like to thank Reviewer 3 for the comments, which will undoubtedly help to improve the manuscript.
We have expanded the introduction of cell entities and considered the manuscript by Yu et al., 2011. The recent new strategies for the detection of CTCs have been discussed in the main manuscript and added to the table 3.
The text of the manuscript has been amended as follows:
Lines 488-490:
“Interestingly, Liu et al. reported that the CTCs present in the 4T1 breast cancer model retained the epithelial phenotype less than the disseminated tumor cells found in the bone marrow [106]”
Lines 494-495:
“Since then, new technical approaches have emerged for the detection of CTC in PE (Table 3). A general review on the approaches of isolation and characterization of CTC can be found elsewhere [108]”
Lines 538-549:
“The preceding identification methodologies are mainly indicated for the detection of big CTCs or EpCAM+/CK+. However, tumors are highly heterogenous and so can be the CTCs, namely highly heterogenous rare CTCs (CRCs) that includes circulating endothelial (CECs) and CTCs. Lin et al proposed an integrative program for the isolation and enrichment of CRCs, which includes subtraction enrichment immunostaining-fluorescence in situ hybridization (SE-iFISH), without hypotonic damage nor anti-EpCAM detection, to enable proper karyotyping for the observation of chromosomal aneuploidy, detection of multiple protein expression and cytogenetic arrangements. Altogether, it can be very informative of the CTC-CEC crosstalk in relation to tumor development processes, such as tumor angiogenesis, drug resistance and metastasis [124]. The same authors identified individual aneuploid CD31+ cells as well as fusion clusters of endothelial-epithelial aneuploid tumor cells with SE-iFISH technology in several biofluids including PE [118].”
New references added to the manuscript
[106] Liu X, Li J, Loureiro Cadilha B, Markota A, Voigt C, Huang Z, Lin PP, Wang DD, Dai J, Kranz G, Krandick A, Libl D, Zitzelsberger H, Zagorski I, Braselmann H, Pan M, Zhu S, Huang Y, Niedermeyer S, Reichel SA, Uhl B, Briukhovetska D, Suárez J, Kobold S, Gires O and Wang H. Epithelial-type systemic breast carcinoma cells with a restricted mesenchymal transition are a major source of metastasis. Science Advances 2019, 5(6): eaav4275.
[108] Yu M, Stott S, Toner M, Maheswaran S, Haber DA. Circulating tumor cells: approaches to isolation and characterization. J Cell Biol 2011, 192 (3):373–382.
[118] Lin PP, Gires O, Wang DD, Li L, Wang H. Comprehensive in situ co-detection of aneuploid circulating endothelial and tumor cells. Scientific Reports 2017, 7:9789.
[124] Lin PP. Aneuploid CTC and CEC. Diagnostics 2018, 8(2), 26.